# Recent Report on the Hydrothermal Growth of LiFePO$_4$ as a Cathode Material

Dimitra Vernardou [1,2]

1 Department of Electrical and Computer Engineering, School of Engineering, Hellenic Mediterranean University, 714 10 Heraklion, Greece; dvernardou@hmu.gr
2 Institute of Emerging Technologies, Hellenic Mediterranean University Center, 714 10 Heraklion, Greece

**Abstract:** Various growth processes have been utilized for the development of lithium iron phosphate including microwave treatment, spray thermal decomposition, sol-gel and the hydrothermal route. However, microwave treatment, spray process and sol-gel suffer from high costs and difficulties in controlling growth parameters. In this review paper, recent synthetic strategies, including the raw materials utilized for the hydrothermal growth of lithium iron phosphate, their effect on the basic characteristics and, as a consequence, the electrochemical performance of cathodes, are reported. The advantages of the hydrothermal process, including high material stability, eco-friendliness, low production costs and material abundance, are explained along with the respective processing parameters, which can be easily tuned to modify lithium iron phosphate characteristics such as structure, morphology and particle size. Specifically, we focus on strategies that were applied in the last three years to improve the performance and electrochemical stability of the cathode utilizing carbon-based materials, N-doped graphene oxide and multi-wall carbon nanotubes (MWCNTs), along with the addition of metallic nanoparticles such as silver. Finally, future perspectives on the hydrothermal process are discussed including the simultaneous growth of powders and solid-state electrodes (i.e., growth of lithium iron phosphate on a rigid substrate) and the improvement in morphology and orientation for its establishment and standardization for the growth of energy storage materials.

**Keywords:** hydrothermal process; raw materials; lithium iron phosphate; carbon materials; cathode; lithium-ion batteries





## 1. Li-Ion Batteries–When All Started

Their history began in 1970 when Stanley Whittingham discovered titanium disulfide (TiS$_2$), which was utilized as a cathode in so-called lithium batteries (LIBs) [1]. In 1980, John Goodenough proposed using a metal oxide such as LiCoO$_2$ instead of a metal sulfide since it could provide higher potential values reaching a value up to 4 V [2]. The first commercially available LIB was accomplished by Akira Yoshino in 1985 with petroleum coke as the anode material instead of reactive Li, making the final structure safe enough to use [3]. Stanley Whittingham, John Goodenough and Akira Yoshino were awarded the Nobel Prize in 2019 for their significant contribution to LIBs. One also cannot neglect Rachid Yazami's role in the development of graphite anodes [4]. LIBs were revolutionary in applications such as portable devices, cell phones and laptops and have provided new perspectives on their utilization in electric vehicles (EVs), with significant improvements in performance since their first employment almost 40 years ago.

## 2. Basic Principles of Li-Ion Batteries

The three main components utilized in LIBs are the anode, cathode and electrolyte. During the discharge process, Li$^+$ flow along the electrolyte and e$^-$ along an electrical circuit from the anode to the cathode (Figure 1a). During the charging process, Li$^+$ are

released from the cathode moving to the anode via the electrolyte and e⁻ via the power source (Figure 1b).

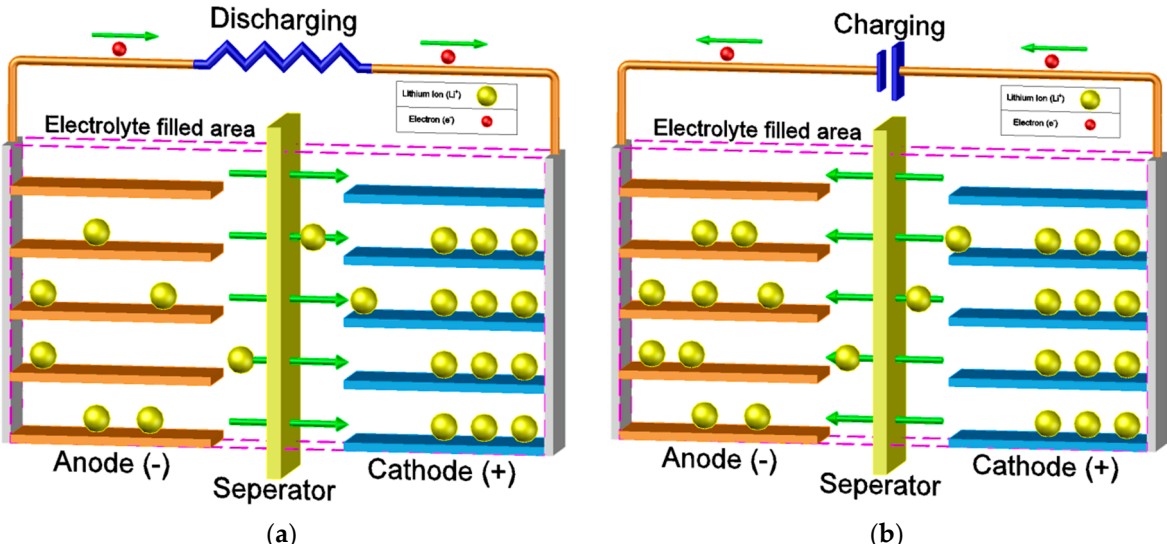

**Figure 1.** A schematic presentation of the discharging (**a**) and charging (**b**) processes in lithium−ion batteries.

Through the years, different cathode materials have been employed to improve battery performance and reduce costs. A single transition metal oxide (TMO) such as lithium cobalt oxide (LCO) is one of the most popular cathodes due to its high energy density, good conductivity, high open-circuit voltage and low self-discharge [5]. However, it is costly, toxic and its resource is no longer abundant [6]. Another option is lithium manganese oxide ($LiMn_2O_4$), which has, however, a lower capacity and less cycle stability than LCO [7]. Another type is the intercalation cathode, which is a solid host network that can store guest ions, such as $LiTiS_2$ (LTS) [8]. In addition, there are conversion cathode materials such as $FeF_2$, LiBr and $Li_2S$-C, which undergo a redox reaction during the $Li^+$ intercalation/de-intercalation that changes their crystalline structure [9,10]. Lithium sulfide has attracted great interest since sulfur has a high theoretical capacity and is one of the most abundant materials on earth, thus, decreasing its cost. However, there are drawbacks to this material such as the shuttle of lithium polysulfides [11]. Ternary compounds with Ni-rich layered oxides such as $LiNi_{0.8}Co_{0.15}Al_{0.05}O_2$ (NCA) and $LiNi_{0.8}Co_{0.1}Mn_{0.1}O_2$ (NCM811) are known as cathode materials because the redox activity of nickel occurs at higher potential vs. $Li^+$/Li than iron in lithium iron phosphate (LFP) [12]. Nevertheless, the increase in nickel concentration results in a decrease in structural stability due to the weaker Ni-O bonds (i.e., lower cycling stability). Furthermore, the chemical reactivity of the surface layer is increased due to the oxidizing $Ni^{3+}$/$Ni^{4+}$ redox potential [12]. High entropy materials have raised attention for use in rechargeable batteries due to their superior $Li^+$ conductivity at room temperature, which allows them to achieve a high and stable specific capacity [13]. A comparison of the electrochemical parameters for some cathode materials is listed in Table 1. It is important to note that cycling $LiCoO_2$ to voltages higher than 4.35 V results in structural instability and capacity fading, exhibiting a maximum capacity of ~165 mAh·g⁻¹ [14].

**Table 1.** Theoretical specific capacity and specific energy values of some cathodes.

| Cathode | $LiFePO_4$ | $LiMn_2O_4$ | $LiCoO_2$ | $Li_2TiS_3$ |
|---|---|---|---|---|
| Specific capacity/mAh·g⁻¹ | 170 [15] | 148 [16] | 274 [17,18] | 339 [19] |
| Specific energy/Wh·kg⁻¹ | 590 [20] | 560 [20] | 980 [20] | 810 [19] |

The $LiFePO_4$ cathode battery is similar to the lithium nickel cobalt aluminum oxide ($LiNiCoAlO_2$) battery; however, it is safer and fairly widely used in automotive and other areas [21]. In addition, $LiFePO_4$ batteries have lower energy than nickel manganese cobalt oxide (NMC)-based batteries; they have a longer lifetime, they are safer and they use cheap and abundant materials.

In particular, iron-based compounds are promising choices due to the abundance, low cost and lower toxicity of Fe as compared to Co, Ni or Mn. The olivine lithium iron phosphate ($LiFePO_4$, LFP) is currently under further study because of its low cost and toxicity, and high specific capacity of 170 mAh·g$^{-1}$ [22]. In LFP, approximately 0.6 lithium atoms can be extracted at a closed-circuit voltage of 3.5 V vs. Li. During the charging process, $Li^+$ are removed from the cathode to give $FePO_4$ (Equation (1)), while in the discharging process, the route is inversed with the insertion of Li into $FePO_4$ (Equation (2)) [16,23].

$$LiFePO_4 + Li^+ + xe^- \rightarrow xFePO_4 + (1 - x)LiFePO_4 \tag{1}$$

$$FePO_4 + xLi^+ + xe^- \rightarrow xLiFePO_4 + (1 - x)FePO_4 \tag{2}$$

It was found that through Alloy-Theoretic Automated Toolkit (ATAT) analysis for olivine $Li_{1-x}FePO_4$ ($0 \leq x \leq 1$), the structural framework is maintained up to 90% of $Li^+$ de-intercalation (corresponding to structure h), with 10 intermediate stable phases being involved during the $Li^+$ intercalation/deintercalation process [24]. The theoretical identification of 10 intermediate stable phases is consistent with reported experimental findings [25]. However, previous studies with density functional theory (DFT) treated the systems with the two terminal phases, resulting in the incomplete prediction of a full voltage profile [24].

## 3. Features of LiFePO$_4$

LFP was first introduced by a team including Goodenough in 1997. It is a polyanion material crystallized in an orthorhombic system with a slightly distorted hexagonally close-packed oxygen arrangement called olivine (Figure 2 [26,27]). It is stabilized through the phosphorous-oxygen bond, which decreases oxygen release during the cycling process [23]. The biggest advantages of LFP are its good thermal and electrochemical stability, environmental friendliness and low cost [8,16]. The good structure stability is due to the phosphate olivine crystal structure, which allows $Li^+$ to diffuse into a 1D tunnel along with its b-axis as shown in Figure 2. On the other side, it has low $Li^+$ diffusion and electronic conductivity, which leads to loss of capacity [27,28].

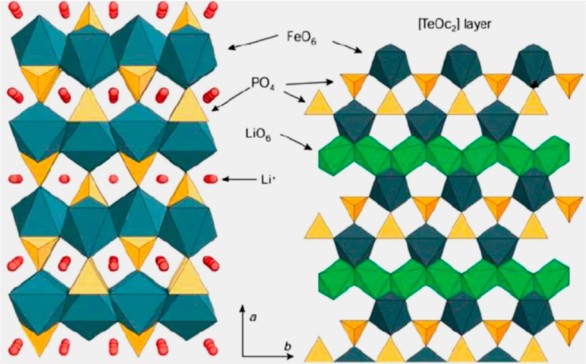

**Figure 2.** The crystal structure of olivine $LiFePO_4$. Adapted with permission from Ref [27]. 2001, Springer Nature.

Various strategies have been performed to improve the electrochemical performance of LFP including particle size reduction [29], ion doping [30], carbon-coating [31], metal nanoparticle deposition [32], metal oxide [33,34] and optimization of the synthetic procedures. The commonly used preparation methods of LFP include microwave treatment [35,36],

spray thermal decomposition [37,38], sol-gel [39,40] and hydrothermal method [41]. These methods can produce high-purity and homogeneous size distribution LFP particles, which can provide improved electrochemical performance [42]. However, microwave treatment, spray thermal decomposition and sol-gel suffer from high costs and difficulties in controlling the reaction (Table 2). Therefore, simplifying the preparation technology to obtain a product with small and uniform distribution remains a challenge. Hydrothermal is an auspicious route because of its simple operation, low reaction temperature, easiness in tuning the morphology and structure of the materials through processing parameters including temperature, time and low-toxicity raw materials [33–46]. Hydrothermal synthesis is referred to as the heterogeneous reaction for synthesizing inorganic materials in aqueous media. In particular, the aqueous mixture of precursors is heated in an autoclave bottle above the boiling point of water and the atmospheric pressure. The water's properties, such as density and dielectric constant, are varied with temperature and pressure, both of which can control the nucleation, allowing the tuning of the crystal phase and particle size [47]. Furthermore, water makes the process compatible with green and sustainable chemistry [47].

**Table 2.** Comparison of growth processes.

| LiFePO$_4$ | Advantages | Disadvantages |
| --- | --- | --- |
| Microwave assisted synthesis | pure products, control over reaction parameters, green raw materials (H$_2$O, alcohols) | expensive equipment unfeasible reaction monitoring |
| Spray pyrolysis | narrow particle size distribution, homogeneous preparation | plethora of parameters to control (solute concentration, temperature, temperature gradients, residence time in furnace and carrier gases) |
| Sol-gel | homogeneous and high adhesion products, low temperature processing | safety matters concerned since countable amounts of by-products are released in calcination step long growth period |
| Hydrothermal method | simple, easy and low-cost method, production of high-quality nanostructures through an easy control of growth parameters | long growth period |

The formation of nanomaterials can occur in a wide temperature range, from room temperature to higher values, permitting the growth of even, flexible substrates. Hydrothermal growth is a crystallization process. In particular, nucleation occurs when the solubility of a solute exceeds the limit in the solution with the subsequent solute precipitation into clusters of crystals. A typical hydrothermal reaction of LFP is the following Equation (3) [48]

$$3LiOH + FeSO_4 + H_3PO_4 \rightarrow LiFePO_4 + Li_2SO_4 + 3H_2O \tag{3}$$

In the following sections, we will present the most recent synthetic hydrothermal strategies of LFP, with emphasis on the raw materials utilized, their basic characterization and their potential utilization as cathodes in LIBs. In addition, we will underline the potential of the hydrothermal route for utilization in solid-state electrodes in saving time and lowering costs. As one can observe in Figure 3, there is an increasing interest in review papers over the last few years, emphasizing the necessity for improvement in cathodes' growth and performance.

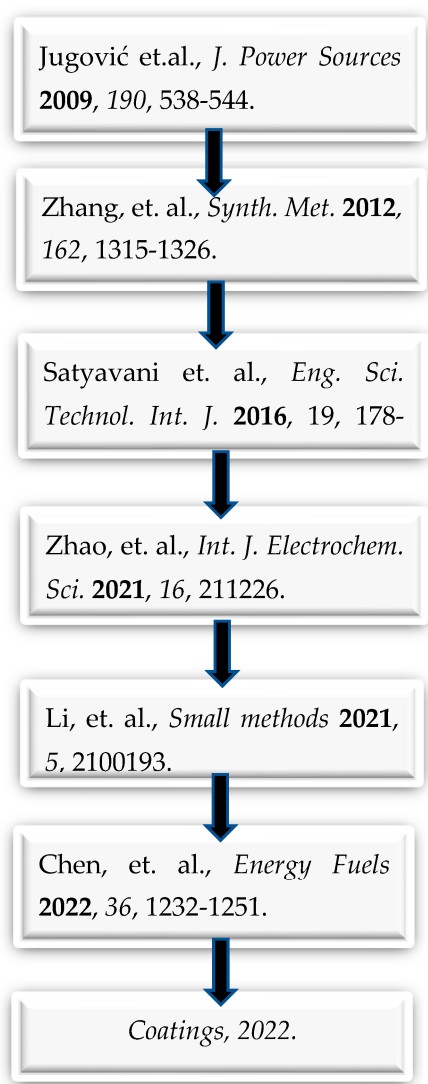

Jugović et.al., *J. Power Sources* **2009**, *190*, 538-544.

Zhang, et. al., *Synth. Met.* **2012**, *162*, 1315-1326.

Satyavani et. al., *Eng. Sci. Technol. Int. J.* **2016**, 19, 178-

Zhao, et. al., *Int. J. Electrochem. Sci.* **2021**, *16*, 211226.

Li, et. al., *Small methods* **2021**, *5*, 2100193.

Chen, et. al., *Energy Fuels* **2022**, *36*, 1232-1251.

*Coatings, 2022.*

**Figure 3.** Time frames of review papers on the synthesis of LiFePO$_4$ [49–54].

*3.1. Hydrothermal Synthesis of LiFePO$_4$*

An approach to increasing the electronic conductivity of LFP and providing paths for the easy insertion of electrons is the utilization of conductive materials such as carbon. Carbon as a coating has the following advantages: a. increases conductivity, b. avoids the further growth of LFP grains and c. prevents oxidation of Fe$^{2+}$ to Fe$^{3+}$ verifying the purity of the material [55,56]. A typical hydrothermal synthesis involves the following reactants: lithium hydroxide monohydrate (LiOH·H$_2$O), iron sulfate heptahydrate (FeSO$_4$·7H$_2$O) and phosphoric acid (H$_3$PO$_4$) [57]. They are all dissolved in deionized water following the stoichiometric amounts of LFP in the Ar atmosphere. After this procedure, they are transferred to autoclavable bottles and heated at 140, 160, 180 and 200 °C for 10 h. In that way, LFP powders are synthesized; they are then further heated at 650 °C for 6 h under an Ar atmosphere. Following this experimental procedure, different hydrothermal reaction times of 6, 8, 10, 12 and 15 h at 180 °C were performed. Finally, a carbon coating process is employed with glucose to make LFP/C composites. X-ray diffraction (XRD) patterns indicated the formation of pure phase LFP material with the only exception of a hydrothermal reaction time of 6 h. Scanning electron microscopy (SEM) presented a better dispersion of LFP particles with the increase in reaction temperature up to 180 °C, while at 200 °C the particles re-aggregated severely.

Based on the above approach, the ratio between the reactants can be varied for Li:Fe:P = 3:1:1 along with the addition of polyethylene glycol 2000 (PEG 2000), polyvinylpyrrolidone (PVP) and cetyltrimethyl ammonium bromide (CTAB) [58]. The last three reactants act as stabilizers and growth modifiers for the morphology optimization and particle refinement of LFP. It was investigated that the particle size of LFP through PEG was uniform, showing a flat rhombohedron-like shape with the length of the large diagonal being about 1.5 μm, the short diagonal about 0.5 μm, and a thickness of about 0.3 μm; these synthesized with PVP presented a porous structure, while those grown using CTAB indicated a flower-like morphology with a diameter of 3~5 μm, which is gathered by single-crystal particles (Figure 4a–d). It can be observed that the addition of surfactants has caused a significant change in the morphology of the final material. Regarding the XRD patterns, the olivine structure with the $P_{nma}$ space group is shown in all cases. The diffraction peaks indicate that the presence of surfactants does not affect the phase of the material (Figure 4e).

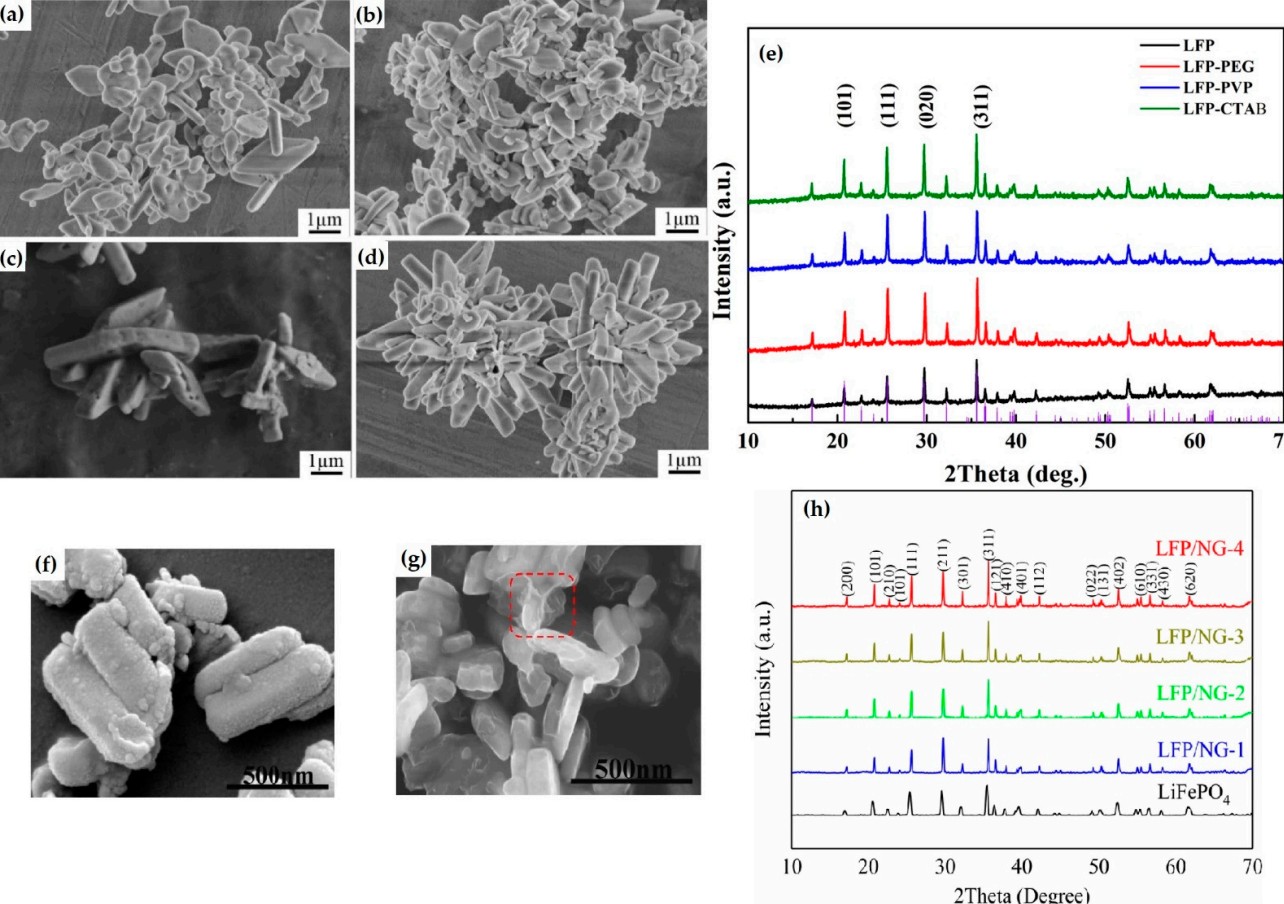

**Figure 4.** SEM images of LiFePO$_4$ (**a**), LiFePO$_4$ with PEG-assisted (**b**), PVP-assisted (**c**) and CTAB-assisted (**d**) and their XRD patterns (**e**) [58]. SEM images of LiFePO$_4$ (**f**) and LiFePO$_4$ with N-doping (**g**) and their XRD patterns (**h**) [59] (Note: NG stands for the ratio of melamine to graphene oxide. NG-1, NG-2, NG-3 and NG-4 stand for 20:1, 15:1, 10:1 and 5:1, respectively). Adapted with permission from Ref [58]. 2021, Springer Nature and Ref [59]. 2021, Elsevier.

Another modification of the procedure involves the addition of N-doped GO, which was prepared using the hummer method [59]. N-doping is introduced to strengthen the conductivity and chemical activity of graphene. This doping procedure is necessary to prevent damage to its electronic conductivity caused by the introduction of oxygen functional groups. In that way, a three-dimensional conductive network structure was synthesized via one–step in situ hydrothermal growth. SEM images presented a slight reduction in the LFP particle size (Figure 4f,g). The size is smaller, more uniform and more

dispersed than in a single LFP. It is also worth noting that LFP particles are well covered and connected by N-doped GO through this one-step in situ hydrothermal growth. XRD indicated that appropriate N doping amounts do not alter the crystal structure of LFP. All peaks are consistent with the olivine $P_{nmb}$ space group. One can observe that there are no diffraction peaks of NG (i.e., NG stands for the ratio of melamine to graphene oxide) or carbon showing that the added NG does not affect the crystal structure of the $LiFePO_4$ material (Figure 4h).

LFP can also be grown using sodium dihydrogen phosphate dehydrate ($NaH_2PO_4$), lithium acetate ($LiOOCCH_3$), ferric nitrate nonahydrate ($Fe(NO_3)_3 \cdot 9H_2O$) and CTAB [26]. The final solution was held in an oven at 180 °C for 6 h. LFP powders were finally dried and annealed at 800 °C. In that case, the orthorhombic olivine structure of the disc form with a particle size distribution in the range of 150–600 nm was obtained.

Furthermore, the controllable synthesis of LFP microparticles and microrods was possible through the modulation of synthetic parameters [60]. In the case of microparticles, $LiOH \cdot H_2O$, iron chloride ($FeCl_2$), $H_3PO_4$ and ascorbic acid were utilized for the solution preparation, which was further heated at 160 °C for 12 h in an oven. For the microrods synthesis, $LiOH \cdot H_2O$, $FeSO_4$, $H_3PO_4$, ascorbic acid and ethylene glycol were employed. This solution was also heated at the same temperature as the previous one. Finally, the precipitates were annealed at 700 °C for 6 h under an Ar atmosphere. Ascorbic acid acted as a reducing agent to prevent the oxidation of $Fe^{2+}$. In both cases, the single-phase particles and rod-shaped morphologies were confirmed by XRD and SEM analysis, respectively.

The improvement of LFP conductivity can be also achieved through the addition of metallic nanoparticles (i.e., Ag) along with the addition of C and rGO [61]. Initially, the LFP/C composite (Figure 5a) was prepared through the mixing of LiOH and a glucose-aqueous solution with the dropwise addition of $FeSO_4$ and $H_3PO_4$. The molar ratio was kept at Li:Fe:P = 3:1:1. In this solution, silver ammonia solution (i.e., obtained after the titration of $NH_3 \cdot H_2O$ with $AgNO_3$ solution) and $CH_3CHO$ were added. The 1D Ag-nanochains (NCs) bridged LFP/C forming a 3D network with the aid of rGO (Figure 5b). The mole ratio of $AgNO_3$ and $NH_3 \cdot H_2O$ was 1:2, while a 5% excess of $CH_3CHO$ was introduced for the $Ag^+$ to be fully reduced. The resulting solution was transferred to a Teflon-lined stainless steel autoclavable and heated at 200 °C for 15 h. The powders were finally annealed at 600 °C for 2 h under an $N_2$ atmosphere. This route is a new method for using 1D metal materials as wires to construct a 3D network and enhance the conductivity of LFP. XRD indicated that all samples are indexed to an olivine phase (Figure 5c).

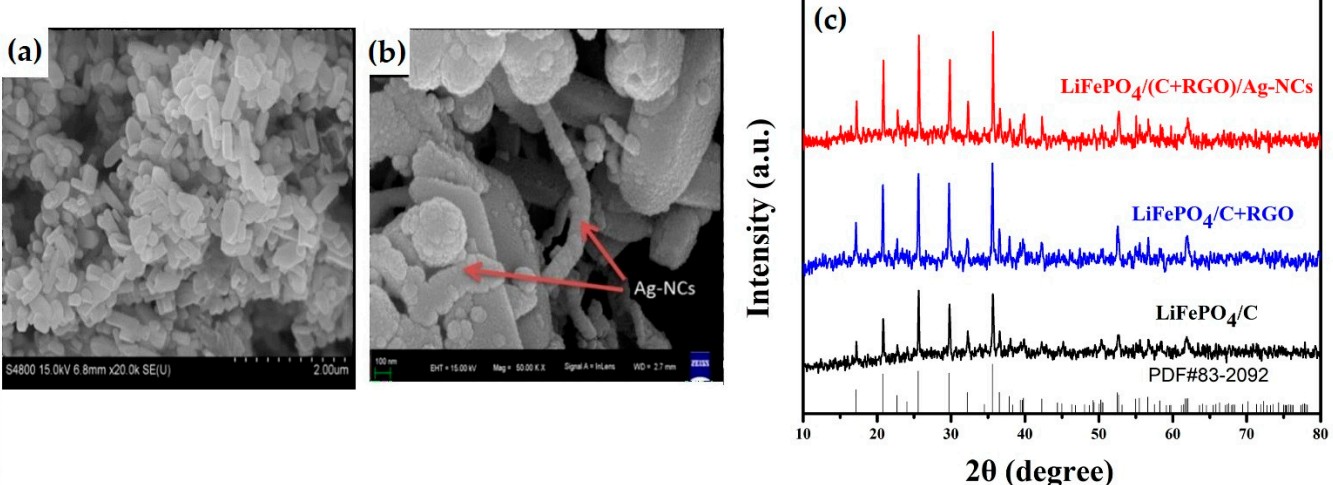

**Figure 5.** SEM images of $LiFePO_4$/C (**a**) and $LiFePO_4$/(C + rGO)/Ag-nanochains (**b**) and their XRD patterns (**c**) [61].

General Remarks

One may say that, in all approaches, the control of morphology and structure can be simply achieved through the tuning of processing parameters (Table 3). For instance, the reaction time is critical to obtaining pure phase LFP material [57], and the addition of stabilizers can alter the morphology of LFP (particle size with PEG, flat rhombohedron-like shape with PVP and flower-like with CTAB) [58], the addition of N-doped GO can increase the conductivity of particles [59], the utilization of other Fe source such as $Fe(NO_3)_3 \cdot 9H_2O$ and $FeCl_2$ is able to result in disc form and microparticles, respectively [56]. In general, the reactants displayed low toxicity. In terms of the processing costs, we could say that it was not expensive, but also not cheap considering the high processing times and temperatures. Finally, it is also worth noting that in all cases, the growth of LFP powders was accomplished.

**Table 3.** Comparison of synthetic processes during hydrothermal conditions.

| LiFePO$_4$ | Precursors | Molar Ratio of Reactants | Fe Source |
|---|---|---|---|
| disc form [26] | LiOOCCH$_3$, Fe(NO$_3$)$_3$·9H$_2$O, NaH$_2$PO$_4$, CTAB | - | Fe(NO$_3$)$_3$·9H$_2$O |
| Nanoparticles [42] | Naphthenic acid, isooctyl alcohol, FeSO$_4$·7H$_2$O, LiOH, H$_3$PO$_4$ | LiOH:H$_3$PO$_4$, 1.2:1 | FeSO$_4$·7H$_2$O |
| Nanoparticles [57] | LiOH·H$_2$O, FeSO$_4$·7H$_2$O, H$_3$PO$_4$, glucose | - | FeSO$_4$·7H$_2$O |
| Flower-like morphology [58] | LiOH·H$_2$O, FeSO$_4$·7H$_2$O, H$_3$PO$_4$, CTAB | LiOH:FeSO$_4$:H$_3$PO$_4$ = 3:1:1 | FeSO$_4$·7H$_2$O |
| Flat rhombohedron-like shape [58] | LiOH·H$_2$O, FeSO$_4$·7H$_2$O, H$_3$PO$_4$, PEG | LiOH:FeSO$_4$:H$_3$PO$_4$ = 3:1:1 | FeSO$_4$·7H$_2$O |
| Porous structure [58] | LiOH·H$_2$O, FeSO$_4$·7H$_2$O, H$_3$PO$_4$, PVP | LiOH:FeSO$_4$:H$_3$PO$_4$ = 3:1:1 | FeSO$_4$·7H$_2$O |
| 3D conductive network structure [59] | LiOH·H$_2$O, FeSO$_4$·7H$_2$O, H$_3$PO$_4$, graphite powder, H$_2$SO$_4$, KMnO$_4$, melamine | - | FeSO$_4$·7H$_2$O |
| Microparticles [60] | LiOH·H$_2$O, FeCl$_2$, H$_3$PO$_4$, ascorbic acid, ethylene glycol | LiOH:FeCl$_2$:H$_3$PO$_4$, 3:1:1 | FeCl$_2$ |
| Microrods [60] | LiOH·H$_2$O, FeSO$_4$, H$_3$PO$_4$, ascorbic acid, ethylene glycol | LiOH:FeSO$_4$:H$_3$PO$_4$, 3:1:1 | FeSO$_4$ |
| 3D conduction network connected by 1D helix-like Ag nanochains [61] | LiOH, FeSO$_4$, H$_3$PO$_4$, NH$_3$·H$_2$O AgNO$_3$, CH$_3$CHO | LiOH: FeSO$_4$: H$_3$PO$_4$, 3:1:1 | FeSO$_4$ |

Another perspective could be the development of solid-state LFP (i.e., direct growth of LFP on a rigid substrate). The advantages of solid-state material are the following: (a) compact in nature, (b) good controllability of the interface between two different materials (i.e., one-step growth) and (c) less expensive (i.e., centrifuge, drying and calcination of the powder are avoided). Nevertheless, the mass production and manufacturing of solid-state electrodes are still in progress. Taking into consideration deposition techniques such as chemical vapor deposition (CVD) or approaches that require templates and, as a consequence, sophisticated equipment and high growth temperatures, the hydrothermal route is more economical and eco-friendly for the preparation of nanostructured materials. To be specific, a substrate can be positioned on the bottom of the autoclavable glass bottles controlling the solution chemistry (i.e., temperature, pH, concentration and molar ratio), trying to avoid organic additives or substrate pre-treatment targeting in a one-step process and widening the type of substrate utilized.

*3.2. Electrochemical Evaluation of LiFePO$_4$*

Based on the hydrothermal routes utilized, it was observed that appropriate alteration of the processing parameters (i.e., temperature and growth period) can control the

structure and the active surface area of the material, obtaining a high reversible capacity of 139.5 mAh·g$^{-1}$ [57]. Taking advantage of the utilization of surfactants to alter the morphology, one can find that PEG 2000 rectifies the particles, shortening the diffusion path for Li$^+$, which is favorable for the intercalation/de-intercalation processes; exhibiting a high initial discharge specific capacity of 122.80 mAh·g$^{-1}$ with a capacity retention rate of 95.50% after 100 cycles at 0.1 C (Figure 6a,b) [58]. The smaller particle size with uniform distribution favors the transmission of the electrolyte, enhancing the electrochemical performance of the materials compared to those grown with CTAB-assisted and PVP-assisted technology. The last two samples also show good cyclic stability but suffer from a large capacity loss in the initial cycle [59]. N-doped graphene was also detrimental to improving the conductivity of the material, reducing the electrode polarization and improving reversibility, presenting a specific capacity of 166.6 mAh·g$^{-1}$ at a rate of 0.2 C (Figure 6c) [59]. The specific capacities continuously increase for higher N-doped graphene content. In that way, the graphene is better reduced, introducing more defects and reactive sites for graphene to enhance its chemical activity and electrical properties. It is noted that the voltage range varied among LiFePO$_4$-PEG (~2.2–4.0 V) and LiFePO$_4$, with different N-doped graphene (~2.5–4.2 V) as one can see in Figure 6b,c, respectively. This behavior may be due to the variation in electrode polarization leading to alterations in the distribution of electrode active material and, as a consequence, the Li$^+$ intercalation on the surface of the electrode. Another research work confirmed the significance of materials morphology on electrochemical performance indicating that a homogeneous morphology of LFP-microrods/multi-wall carbon nanotubes can shorten the Li$^+$ diffusion path and decrease internal resistance improving the electrochemical reversibility during the intercalation/de-intercalation processes [60]. In particular, Figure 6d shows the galvanostatic charge-discharge curves of LFP-microparticles/multi-wall carbon nanotubes (MWCNTs) and LFP-microrods/MWCNTs at 0.1 C for 25 cycles. Their initial discharge capacity was approximately 159 and 192 mAh·g$^{-1}$ for LFP-microparticles/MWCNT and LFP-microrods/MWCNT, respectively. We need to note that the discharge capacity is higher than the charge capacity, which may be due to irreversible losses in the charge capacity owing possibly to the formation of solid electrolyte interfaces and/or irreversible captured Li$^+$ on Cu surfaces, resulting in the formation of lithium oxides. The charge capacity (coming from lithium removal from the LiFePO$_4$ structure) is also too high. In that case, we believe that copper should be avoided. Furthermore, the capacity retention for LFP-microrods/MWCNTs is higher (approximately 83%) compared to LFP-microparticles/MWCNTs (approximately 66%) confirming good electrochemical stability with lesser degradation [60]. Excellent structure stability and electrochemical performance can also be obtained through a 3D conductive network formed by silver nanochains (NCs) and rGO, which play an important role as stabilizers during the intercalation/de-intercalation processes, preventing structural collapse and ensuring the integrity of the material [61]. Moreover, the graphene acts as a storage for Li$^+$ (i.e., more Li$^+$ are embedded and released in a cycle, gradually increasing the specific capacity) [62]. Interestingly, one can notice the stability of LiFePO$_4$/C, LiFePO$_4$/C + rGO and LiFePO$_4$/(C + rGO)/Ag-NCs at the 0.2 C rate in Figure 6e. It is presented that LiFePO$_4$/(C + rGO)/Ag-NCs has the highest specific capacity with the rest being 150.7 mAh·g$^{-1}$. Nevertheless, fluctuations are observed above 10 cycles due to side reactions occurring at a particular time period. Further measurements are required to understand the particular behavior, including Nyquist plots. Following a simpler route for the hydrothermal synthesis of LFP through the pH, temperature and time adjustment, different morphologies and particle sizes can be grown [42]. In this work, it was found that nanoparticles with regular morphology and small size have a high discharge capacity of 156.1 mAh·g$^{-1}$ at 0.1 C after 40 cycles (Figure 6f).

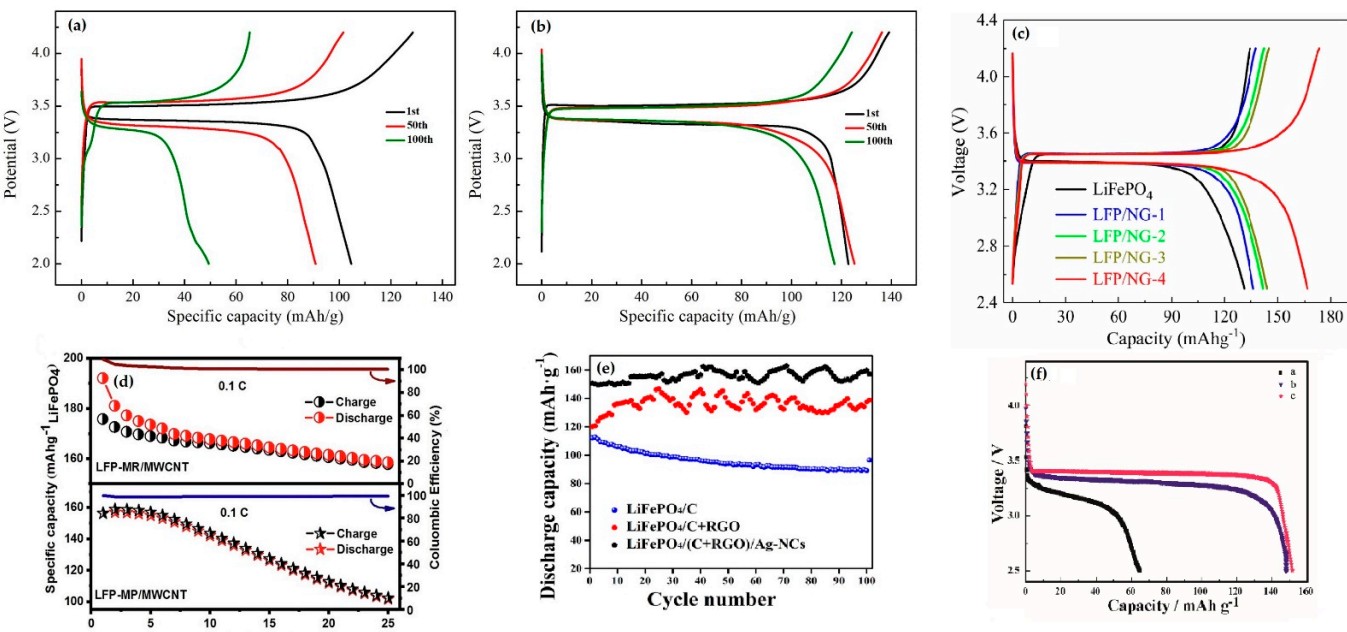

**Figure 6.** Discharge/charge profiles for different cycle numbers of LiFePO$_4$ (**a**) and LiFePO$_4$-PEG (**b**) [58]. First charge/discharge curves of LiFePO$_4$ with different N-doping values (**c**) [59]. Discharge/charge curves of LiFePO$_4$-microrods/MWCNTs (**up**) and LiFePO$_4$-microparticles/MWCNTs (**down**) (**d**) [60]. Specific capacity profiles during charge/discharge of LiFePO$_4$/C, LiFePO$_4$/C + RGO, LiFePO$_4$/(C + RGO)/Ag-NCs at 0.2 C rate for 100 cycle numbers (**e**) [62]. First discharge/charge profiles of LiFePO$_4$ with blocky–a, rod-shaped–b and rod-shaped–c morphology. The first one was grown at 160 °C, while the other two were at 220 and 250 °C, respectively (**f**) [42]. Adapted with permission from Ref [58]. 2021, Springer Nature; Ref [59]. 2021, Elsevier; Ref [60]. 2021, Elsevier and Ref [62]. 2013, Springer Nature.

From the above, one may conclude that the synergy of suitable morphology with the carbon material (i.e., MWCNTs) is necessary to obtain a high specific capacity with good cycling stability. Others also reported the significant role of MWCNTs generated from the three-phase hybrid MWCNTs-V$_2$O$_5$, which offers a large active surface area, a good conducting network and effective strain upon cycling [63]. This happens because MWCNTs favor the capacity of the material, while the controllable morphology (i.e., large specific surface area) is vital for the enhancement of physical and electrochemical properties (i.e., effective conducting of network buffering against the strain upon cycling).

In Table 4, one can observe a comparison of cathode materials grown by different synthetic routes. The carbon sources are either organic (such as glucose [64], citric acid, oleic acid [65], polydopamine [66,67], ascorbic acid [60], ethylene glycol [68]) or inorganic (such as acetylene black, super P, carbon nanotubes [69] and graphene [56,64,69–71]) precursors. Organic compounds offer uniform thickness, full coverage and homogeneity with however difficulty in conductivity and graphitized degree control. On the other side, inorganic carbon sources present the opposite advantages and disadvantages [49,50,72]. It seems that a combination of organic and inorganic sources can be promising for high-performance cathodes with extra care needed to find the optimum amount of carbon as this can vary with the shape, size and structure of LFP electrodes [73–75]. Based on this consideration, LFPNR@N-C@RGO presented the highest specific capacity and capacity retention over 1000 cycles. In that case, the interior N-C coating enhanced the conductivity of the LFP nanorods (NR) and the exterior GO coating acted as a conducting network to electrically connect the entire electrode. Another interesting material is the LFP-microrods/MWCNT grown through the hydrothermal method, which presented homogeneous coverage of the conductive MWCNT network, thereby shortening the Li$^+$ diffusion path and improving

the electrochemical reversibility during Li$^+$ intercalation/de-intercalation. This particular cathode material combines cost-effectiveness with excellent performance.

**Table 4.** Comparison of electrochemical performance of cathode materials using different growth methods.

| Cathode Materials | Synthesis Process | Specific Capacity (mAh·g$^{-1}$) | Capacity Retention |
|---|---|---|---|
| LFP/GO [56] | Solution combustion/colloidal | 162 at 0.1 C | 96% after 40 cycles at 0.2 C |
| LFP-microrods/MWCNT [60] | Hydrothermal method | 192 at 0.1 C | ~97% after 600 cycles at 10 C |
| LFP@C/G [64] | Rheological phase/solid state | 163.8 at 0.1 C | 92% after 500 cycles at 10 C |
| C-L$_{1.05}$FP [65] | Sol-gel | 155 at C/30 | Excellent cycling stability after 100 cycles |
| N-C@LFP [66] | Hydrothermal plus chemical polymerization | 162.1 at 1 C | 100% after 100 cycles at 10 C |
| LFP/CN [67] | Microwave heating route | 160 at 0.2 C | 97.9% after 50 cycles at 0.1 C |
| LFP NR@N-C@RGO [68] | Surfactant-assisted synthesis | 172 at 0.1 C | 95.8% after 1000 cycles at 10 C |
| LFP-CNT-G [69] | Solid state | 168.9 at 0.2 C | 98% after 100 cycles at 0.2 C |
| LFP@G [70] | Solvothermal/freeze-drying | 163 at 0.2 C | 99.8% after 600 cycles at 10 C |
| LFP/G [71] | Solid state | 161 at 0.1 C | 70 mAh·g$^{-1}$ after 44 cycles at 50 C |

We need to note that the choice of raw materials with reduced environmental risks is important, and presents a challenge for the researchers to design cost-competitive products using green solvents, dry media and energy-efficient synthesis.

## 4. Conclusions

LiFePO$_4$ is a promising cathode material for lithium-ion batteries due to its low cost, low toxicity and properties. Even though it is a well-established technology on the market, there is still space for improvement regarding the growth process. We could say that the hydrothermal-based method is the best approach due to its simplicity and decent cost, with its most important advantage being the simplicity of tuning the structure and morphology of the material. These tuning characteristics can be achieved through the alteration of the processing parameters, the addition of stabilizers and the introduction of metallic nanoparticles and/or carbon coatings. These characteristics can directly affect the electrochemical performance including capacity and stability, thereby reaching a higher theoretical value for the LFP-microrods/MWCNTs (i.e., 192 mAh·g$^{-1}$) compared with the single LFP. The most special aspect of MWCNTs is the 3D nanoarchitecture, which offers better charge transfer capability, mesoporosity, stability and electrolyte accessibility compared to other carbon nanomaterials [76].

We may say that there are many approaches to hydrothermal synthesis including solid-state growth, meaning the direct growth of LFP on a rigid substrate. This approach has low costs, good controllability of the interface between two materials and the material used is compact in nature; however, more research is needed before the standardization of this route for solid-state growth. The great benefit of this particular route is the ability to control grain size, crystalline phase, particle morphology and surface chemistry through adjustment of the solution composition, temperature, solvent properties and additives. It is therefore important to minimize the utilization of surfactants, binders and organic reactants to keep the costs and levels of toxicity as low as possible.

There is also further work that can be done for the improvement in morphology and orientation controls in order to achieve and exceed the theoretical capacity of 170 mAh·g$^{-1}$. Possible directions may include particle size reduction to optimize the Li$^+$ diffusion path

and hybrid coatings (giving priority to inorganic materials to avoid any environmental hazards that may arise), which may enhance the interfacial contact between electrolyte and LFP surface offering more active sites for Li storage.

In the end, the extension of LiFePO$_4$ to other olivine families such as LiMnPO$_4$, LiCoPO$_4$ and LiNiPO$_4$ due to their high operating voltage (4.1, 4.8 and 5.1 V vs. Li$^+$/Li, respectively) [49] is a research challenge for the future.

**Funding:** This research received no external funding.

**Institutional Review Board Statement:** Not applicable.

**Informed Consent Statement:** Not applicable.

**Data Availability Statement:** Not applicable.

**Conflicts of Interest:** The author declares no conflict of interest.

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
