# Peer review of "Recent Report on the Hydrothermal Growth of LiFePO4 as a Cathode Material"

_coatings, doi:10.3390/coatings12101543_

Round 1
Reviewer 1 Report
- The review “Perspectives and Recent Hydrothermal Synthetic Strategies of 2 LiFePO4 as Cathode Material” by Dimitra Vernardou is dealing with recent synthetic strategies for the hydrothermal growth of lithium iron phosphate and their influence on the electrochemical performance of the cathodes. Special attention was payed to the improving the performance and electrochemical stability of the cathode employing carbon based materials, including multi-walled carbon nanotubes. However, there are some issues which should be clarified.
- The most accepted term for MWCNTs, which can be find in the literature is multi-walled carbon nanotubes. It is not clear from the manuscript why specifically the MWCNTs are used as battery cathode materials. For example, in Lines 334 – 337 “From the above, one may conclude that the synergy of suitable morphology with the carbon material (i.e. MWCNTs) is necessary to obtain a high specific capacity with good cycling stability. This happens because carbon favors the capacity of the material, while the controllable morphology (i.e. size and shape) is vital for the enhancement of physical and electrochemical properties”. Possibly carbon nanoparticles or other carbon allotropes aside from MWCNTs, can be used for this purpose. I recommend providing more literature reference concerning utilizing the carbon nanotubes as cathode material and giving details with regard of the MWCNTs and why their using may improve the battery performance. Also, this issue can be emphasized in Conclusions.
- Generally, the review is well-written and comprehensive and can be published in Coatings after minor revision.
Author Response
Dear Reviewer 1,
Thank you very much for your response and helpful suggestions regarding the manuscript entitled:
“Perspectives and Recent Hydrothermal Synthetic Strategies of LiFePO4 as Cathode Material”
I wish to publish in Coatings under the Special Issue of Recent Advancement in Thin Film Deposition, Characterization, and Surface Engineering. I have done major revisions in my original manuscript and I am resubmitting my work hoping that I have fully complied with your recommendations. The highlighted revision encloses all changes in bold and underline.
Reviewer’s comments:
The review “Perspectives and Recent Hydrothermal Synthetic Strategies of LiFePO4 as Cathode Material” by Dimitra Vernardou is dealing with recent synthetic strategies for the hydrothermal growth of lithium iron phosphate and their influence on the electrochemical performance of the cathodes. Special attention was payed to the improving the performance and electrochemical stability of the cathode employing carbon based materials, including multi-walled carbon nanotubes. However, there are some issues which should be clarified.
Thank you very much for suggestions. Please see below my response to your comments.
The most accepted term for MWCNTs, which can be find in the literature is multi-walled carbon nanotubes. It is not clear from the manuscript why specifically the MWCNTs are used as battery cathode materials. For example, in Lines 334 – 337 “From the above, one may conclude that the synergy of suitable morphology with the carbon material (i.e. MWCNTs) is necessary to obtain a high specific capacity with good cycling stability. This happens because carbon favors the capacity of the material, while the controllable morphology (i.e. size and shape) is vital for the enhancement of physical and electrochemical properties”. Possibly carbon nanoparticles or other carbon allotropes aside from MWCNTs, can be used for this purpose. I recommend providing more literature reference concerning utilizing the carbon nanotubes as cathode material and giving details with regard of the MWCNTs and why their using may improve the battery performance. Also, this issue can be emphasized in Conclusions.
Further discussion and references are included in the revised manuscript.
With my best regards,
Dimitra Vernardou, PhD and co-author
Reviewer 2 Report
- The paper titled “Perspectives and Recent Hydrothermal Synthetic Strategies of LiFePO4 as Cathode Material” present the history, the recent advances, and the perspectives of LiFePO4 as the cathode via the hydrothermal strategies. Some words and sentences need to be improved. Some basic physical and chemical problems are not clarified and expressed correctly. Also, this paper needs to be modified to meet what the title interprets, or the title must be modified. I am not agreed this paper to be published.
- In the first sentence, in the abstract, the word of deposition is not appropriate. The word of deposition is often used in lithium metal batteries.
- The advantage of hydrothermal need to emphasize. If this kind of technique is useful in the produce of LFP, thus, the perspectives of hydrothermal is more important.
- In line 15 and 16, the author used ‘low production costs’ for hydrothermal synthetic. As far as I know, the cost is not low.
- Line 29, in 1970 Stanley used TiS2 as cathode and the lithium metal as the anode. This is the so-called lithium batteries not lithium-ion batteries. Also, the paper cited by the author was published in 1975.
- Line 36 and 37, in the development of lithium-ion batteries, Yamani has done a lot of work, which facilitate the development of LIBs. But the reference needs to be the original work of this contribution instead of the review paper.
- Figure 1, the moving way of Li+ and electron is not right. The author needs to learn more about the basic physics of batteries.
- In Table 1, LiCoO2 can only de-intercalate half of it’s lithium, which makes the capacity of ~140.
- In the conclusion part, the author needs to make conclusions. Also, the perspective part is not enough.
Author Response
Dear Reviewer 2,
Thank you very much for your response and helpful suggestions regarding the manuscript entitled:
“Perspectives and Recent Hydrothermal Synthetic Strategies of LiFePO4 as Cathode Material”
I wish to publish in Coatings under the Special Issue of Recent Advancement in Thin Film Deposition, Characterization, and Surface Engineering. I have done major revisions in my original manuscript and I am resubmitting my work hoping that I have fully complied with your recommendations. The highlighted revision encloses all changes in bold and underline.
Reviewer’s comments:
The paper titled “Perspectives and Recent Hydrothermal Synthetic Strategies of LiFePO4 as Cathode Material” present the history, the recent advances, and the perspectives of LiFePO4 as the cathode via the hydrothermal strategies. Some words and sentences need to be improved. Some basic physical and chemical problems are not clarified and expressed correctly. Also, this paper needs to be modified to meet what the title interprets, or the title must be modified. I am not agreed this paper to be published.
Thank you for your suggestions. Please see below our response to your comments.
In the first sentence, in the abstract, the word of deposition is not appropriate. The word of deposition is often used in lithium metal batteries.
The word “deposition” has been replaced.
The advantage of hydrothermal need to emphasize. If this kind of technique is useful in the produce of LFP, thus, the perspectives of hydrothermal is more important.
The advantages and perspectives of hydrothermal growth have been discussed in Abstract, 3. Features of LiFePO4, 3.1.1. General Remarks and 4. Conclusions.
In line 15 and 16, the author used ‘low production costs’ for hydrothermal synthetic. As far as I know, the cost is not low.
I agree with the Reviewer that the hydrothermal growth of LiFePO4 is not low as compared in the basic principle of the process. In Abstract, I refer to the low cost of the process as described in 3. Features LiFePO4.
Line 29, in 1970 Stanley used TiS2 as cathode and the lithium metal as the anode. This is the so-called lithium batteries not lithium-ion batteries. Also, the paper cited by the author was published in 1975.
Unfortunately, it was not possible to find a publication on TiS2 in 1970.
The word “lithium-ion batteries” has been replaced with “so-called lithium batteries”.
Line 36 and 37, in the development of lithium-ion batteries, Yamani has done a lot of work, which facilitate the development of LIBs. But the reference needs to be the original work of this contribution instead of the review paper.
Reference 4 has been replaced accordingly.
Figure 1, the moving way of Li+ and electron is not right. The author needs to learn more about the basic physics of batteries.
I am not quite sure what the mistake is (https://www.linkedin.com/pulse/working-principle-structure-lithium-ion-battery-davide-laverga).
In Table 1, LiCoO2 can only de-intercalate half of it’s lithium, which makes the capacity of ~140.
Related comment and reference have been included before Table 1.
In the conclusion part, the author needs to make conclusions. Also, the perspective part is not enough.
Additional information has been included in Conclusions.
With my best regards,
Dimitra Vernardou, PhD and co-author
Reviewer 3 Report
In this review paper, recent synthetic strategies including the raw materials utilized for the hydrothermal growth of lithium iron phosphate, their effect on the basic characteristics and as a consequence the electrochemical performance of the cathodes, are reported. It is recommended to publish it in this journal after revision.
1、The content of the article is not deep enough, and the latest synthesis method and lithium ion transport mechanism of lithium iron phosphate are not discussed clearly. It is recommended that the author check the latest articles related to cathode. [ 1、J. Energy. Chem, 2021, 59, 229-241 2、https://doi.org/10.1002/eem2.12282 3、Int. J. Miner. Metall. Mater., 28(2021) (2) 305-316 4、ACS applied materials & interfaces, 2019, 11(35): 31991-31996.]
2、Coating the cathode material can improve its conductivity, but on the other hand, it also affects the overall energy density of the battery. How to calculate the mass of the active material of the cathode material after coating?
3、For the common commercial cathode materials LiCoO2, LiFeO4, Nickel-based Cathodes, the author thinks which one has the most application prospects.
4、There are many grammar issues throughout manuscript.
Author Response
Dear Reviewer 3,
Thank you very much for your response and helpful suggestions regarding the manuscript entitled:
“Perspectives and Recent Hydrothermal Synthetic Strategies of LiFePO4 as Cathode Material”
I wish to publish in Coatings under the Special Issue of Recent Advancement in Thin Film Deposition, Characterization, and Surface Engineering. I have done major revisions in my original manuscript and I am resubmitting my work hoping that I have fully complied with your recommendations. The highlighted revision encloses all changes in bold and underline.
Reviewer’s comments:
In this review paper, recent synthetic strategies including the raw materials utilized for the hydrothermal growth of lithium iron phosphate, their effect on the basic characteristics and as a consequence the electrochemical performance of the cathodes, are reported. It is recommended to publish it in this journal after revision.
Thank you very much for your suggestions. Please see below our response to your comments.
The content of the article is not deep enough, and the latest synthesis method and lithium ion transport mechanism of lithium iron phosphate are not discussed clearly. It is recommended that the author check the latest articles related to cathode. [ 1、J. Energy. Chem, 2021, 59, 229-241 2、https://doi.org/10.1002/eem2.12282 3、Int. J. Miner. Metall. Mater., 28(2021) (2) 305-316 4、ACS applied materials & interfaces, 2019, 11(35): 31991-31996.]
The title of the paper has been changed and further discussion is included regarding the Li+ charging/discharging processes.
Note: Only the paper J. Energy. Chem, 2021, 59, 229-241 is related with the material discussed in the particular work.
Coating the cathode material can improve its conductivity, but on the other hand, it also affects the overall energy density of the battery. How to calculate the mass of the active material of the cathode material after coating?
In the case of anode (voltage range between 0-2 V), we need to consider the carbon coating as the part of active material or we need to check capacity contribution from conductive carbon alone and subtract capacity contribution from the anode.
However, in the case of cathode material where the working potential is more than 2 – 2.5 to 4.5 V, we need to consider only the active material weight for capacity calculation.
For the common commercial cathode materials LiCoO2, LiFeO4, Nickel-based Cathodes, the author thinks which one has the most application prospects.
Related discussion is included in the revised manuscript in 2. Basic Principles of Li-ion Batteries.
There are many grammar issues throughout manuscript.
The manuscript has been thoroughly checked by a native speaker prior the submission in the journal.
With my best regards,
Dimitra Vernardou, PhD and co-author
Round 2
Reviewer 2 Report
After the modifications. I think this paper can be published.